# Hydrologic Impacts of Land Use Changes in the Sabor River Basin: A Historical View and Future Perspectives

**Regina Maria Bessa Santos** [1], **Luís Filipe Sanches Fernandes** [1] , **Rui Manuel Vitor Cortes** [1] and **Fernando António Leal Pacheco** [2,*]

[1]   Centre for the Research and Technology of Agro-Environment and Biological Sciences,
     University of Trás-os-Montes and Alto Douro, 5001-801 Vila Real Ap. 1013, Portugal
[2]   Chemistry Research Centre, University of Trás-os-Montes and Alto Douro,
     5001-801 Vila Real Ap. 1013, Portugal
*   Correspondence: fpacheco@utad.pt

**Abstract:** The study area used for this study was the Sabor river basin (located in the Northeast of Portugal), which is composed mostly for agroforestry. The objectives were to analyze the spatiotemporal dynamics of hydrological services that occurred due to land use changes between 1990 and 2008 and to consider two scenarios for the year 2045. The scenarios were, firstly, afforestation projection, proposed by the Regional Plan for Forest Management, and secondly, wildfires that will affect 32% of the basin area. In this work, SWAT (Soil and Water Assessment Tool) was used to simulate the provision of hydrological services, namely water quantity, being calibrated for daily discharge. The calibration and validation showed a good agreement for discharge with coefficients of determination of 0.63 and 0.8 respectively. The land use changes and the afforestation scenario showed decreases in water yield, surface flow, and groundwater flow and increases in evapotranspiration and lateral flow. The wildfire scenario, contrary to the afforestation scenario, showed an increase in surface flow and a decrease in lateral flow. The Land Use and Land Cover (LULC) changes in 2000 and 2006 showed average decreases in the water yield of 91 and 52 mm·year$^{-1}$, respectively. The decrease in water yield was greater in the afforestation scenario than in the wildfires scenario mainly in winter months. In the afforestation scenario, the large decrease varied between 28 hm$^3$·year$^{-1}$ in October and 62 hm$^3$·year$^{-1}$ in January, while in the wildfires scenario, the decrease was somewhat smaller, varying between 15 hm$^3$·year$^{-1}$ in October and 49 hm$^3$·year$^{-1}$ in January.

**Keywords:** SWAT; water balance components; Land Use and Land Cover changes; wildfires; afforestation

## 1. Introduction

Land Use and Land Cover (LULC) are considered the most critical factors affecting the intensity and frequency of surface flow, as well as soil erosion and the loss of nutrients [1,2]. Watershed-level studies have indicated that rapid LULC changes could have significant impacts on water resources [3] and were reported to be an essential factor for controlling water resources on local and global scales during the last century [4,5]. The factors responsible for land use changes are population density fluctuations, changes in the agricultural policy, and the conditions of national and international markets. The main changes occurred due to the abandonment of farmland in less productive mountain areas, the expansion of some subsidized crops to marginal lands, and intense soil erosion during extreme rainstorm events.

Several studies developed in the Mediterranean region showed that inadequate land use and soil cover accelerate water erosion processes and, consequently, lead to land degradation [1,2,6]. Soil erosion is one of the leading causes of the reduction of water quality due to the amount of sediment that arrives at the watercourses and reservoirs [1]. However, many authors have demonstrated that both runoff and sediment loss decrease exponentially as the percentage of vegetation cover increases [1,6]. Thus, forests are often used as a management strategy to improve the provision of ecosystem services in watersheds, because they have a substantial widespread positive influence on climate, hydrology, soils, and biodiversity [7–10].

The studies developed in the 19th century, based on the too-hasty generalization of single point observation, believed that natural and planted forests increased total flow and base flow [3]. Nowadays, studies prove that forests have an impact on the water balance at the basin scale, as forest water consumption is generally higher than that of other vegetation types [3,10,11]. The rise in shrub and forest cover produces declines in water yield, surface, and groundwater flows, and an increase in transpiration. The decrease in the groundwater flow in forests can be justified by the fact that trees with deep roots and high transpiration rates may act as pumps that remove water from the soil and return it to the atmosphere [1,11,12].

In the Mediterranean, erosion is the consequence of complex interactions between environmental and human-related factors. The erosion processes are products of the occurrence of intense rainstorms and long-lasting droughts, high evapotranspiration, the presence of steep slopes, topographic diversity, and recent tectonic activity as well as the recurrent use of fire, deforestation, overgrazing, farming, and construction activities [1,2]. In agreement with the CORINE program, Spain and Portugal are the Mediterranean countries in the European Union facing the highest risk of erosion [13,14]. CORINE means "coordination of information on the environment", and one of the priorities of the program is the evaluation of natural resources and environmental problems in the southern part of the European Community (e.g., soil erosion, water resources, land cover, and coastal problems) [14]. In Portugal, areas of high erosion risk cover almost one-third of the country [15]. For this reason, soil erosion is one of the most intensively studied issues in the Mediterranean region, Portugal included [1,2,6,15]. Erosion and land degradation became a problem in Portugal when arable farming expanded into marginal areas, namely cereal cultivation until the middle of the twentieth century [1]. The introduction of modern agriculture led to the abandonment of traditional or semi-traditional agriculture in mountainous areas as well as in areas with difficult access, resulting in fundamental transformations to the landscape, characterized by the spread of natural vegetation, including both shrubland and forestland [1]. The natural afforestation has resulted in a decline in water resources and surface flow and has decreased soil loss and sediment delivery, as well as caused a progressive improvement in soil characteristics [1]. It has been extensively documented that the rise in shrub and forest cover has promoted lower soil losses and sediment yields than in arable land [2,6].

However, wildfires have been responsible for sudden increases in erosion rates, and they can also result in land degradation and sometimes desertification over the long-term [16]. The occurrence of forest fires affecting thousands of hectares each year is a significant problem in the Mediterranean basin [2,16]. In Portugal, fires are essentially human-caused, and their extension and severity are dependent on extreme weather [17]. In central Portugal, at Pedrógão Grande-Góis, a tragic wildfire occurred on June 17, 2017, with an official death toll of 64 people, almost 500 buildings destroyed, and a continuous patch of more than 42 thousand hectares burned in one week [18]. The areas burned correlated well with socioeconomic and environmental characteristics (e.g., population density and land use, climate, weather, topography, and vegetation cover) [16,17]. However, in this tragic fire, the climate and meteorology played key roles in the initiation and spreading of the wildfires [18]. The warmer and drier than average spring made the landscape prone to the occurrence of large fires and extreme weather conditions favored the ignition and spread of wildfires [18].

It is generally accepted that fires increase runoff, soil erosion, and nutrient and pesticide transport to the river [16,19]. In the Mediterranean region, the most substantial erosion rates and nutrient

losses tend to occur during the first rainstorms after wildfire occurrence during the dry season. The increases in runoff are due to the reduced infiltration capacity of the soil caused by the removal of vegetation and soil organic matter by fire [16]. Wildfires lead to the release of nutrients through the combustion of vegetation and exposure of the soil to erosive factors. Consequently, wildfires are responsible for hydrologic problems and the degradation of water quality caused by excessive input of nutrients, such as phosphorus and nitrogen [16,20]. The excess nutrients affect primary production and, consequently, the eutrophication of aquatic systems. The relationship between the risk of eutrophication and nutrient exports from burned areas has been widely documented [16,19,20].

Hydrological models are computational tools that perform a mathematical representation of hydrological processes, such as infiltration of water into the soil, recharge of aquifers, runoff and drainage network flow [21,22], as well as hydrochemical processes such as weathering or contaminant transport [23–31]. They are also the basis of decision support systems, which can help watershed managers in the control of extreme events such as floods [32,33], or in the assessment of water resource availability [34–40] and threats to water quality [41–48]. The SWAT (Soil and Water Assessment Tool) was the hydrological model used to model the physical processes that occurred in the Sabor river basin. The SWAT model is one of the most widely used water quality watershed and river basin scale models worldwide [49]. It has been applied from small hydrographic basins to the continental scale. Some examples of applications are in North America [50], Europe [51], and Australia [52,53]. These studies were done in four main categories: hydrologic modelling, sediment transport, nutrient and pesticide transport, and scenario analyses.

Within this framework, the following three demands were addressed: (i) establishment of the hydrologic baseline with calibration over a 39-year period (1960–1999), (ii) assessment of the land cover and land use changes between 1960 and 2008 and their effects on water balance components, and (iii) assessment of the forecast of the afforestation and wildfire scenarios for 2045, also on water balance components. A majority of articles discussing environmental effects of land use change and dealing with scenario creation only assess some water balance components, namely water yield or surface flow. This work intends to study the changes occurring not only in water yield or surface flow but also in evapotranspiration, lateral flow, and in groundwater flow.

## 2. Materials and Methods

### 2.1. Study Area

The Sabor river basin is located in the northeast of Portugal and drains into the International Douro basin (Figure 1). The Sabor river has an extension of 212.6 km. Its source is located in Spain at an altitude of about 1600 m, and its mouth is located in the Douro River at about 88 m [54]. The Sabor river basin covers an area of about 3834.5 km$^2$, of which 3170.7 km$^2$ is in Portugal. The average slope of the basin is 16.2% according to the digital elevation model [54]. The slope below 10% is located in the upper zones and plateaus of the basin, and the most rugged slope is located on the scarps and banks of the Sabor river and its tributaries. The main tributaries are the Maças river (drainage area: 720 km$^2$) and the Angueira river (drainage area: 540 km$^2$). In the Sabor river, the Sabor hydroelectric system was built. It comprises two retention dams, known as upstream and downstream walls, which are located at 12.6 and 3 km from the mouth of the Sabor river, respectively [55].

The climate is close to the Mediterranean type. It is characterized by warm-dry summers and precipitation concentrated in the winter and spring seasons. The average annual basin values of rainfall and evapotranspiration are approximately 730 and 540 mm·year$^{-1}$, respectively [56].

The soil characterization in the Sabor river basin is mainly lithosols (87% of the catchment area), but also with the presence of cambisols (7.4%), alisols (3%), anthrosols (1.4%), and fluvisols (0.7%) [57]. The 1990 CORINE Land Cover data published by the European Environmental Agency [58] identified the area as having 59% agricultural areas, 32% semi-natural areas, 9% forest, and less than 1% artificial areas and water bodies (Figure 1b). Agriculture and livestock farms are the main economic activities

in the region, with olive and almond being the main crops [59]. According to the 2011 demographic census, in the Sabor river basin, the population density was 20 inhabitants per km$^2$ [60].

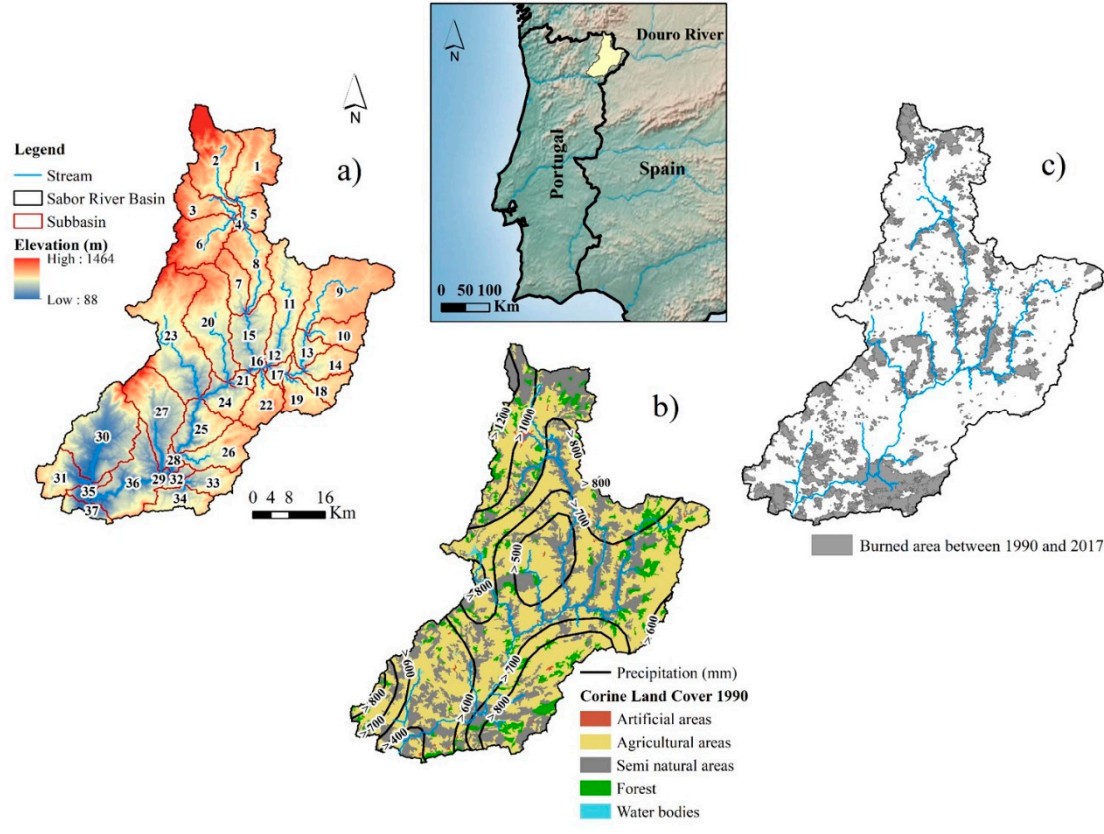

**Figure 1.** The spatial distribution of (**a**) the drainage network, the topography, and the sub-basins of the Sabor river basin; (**b**) the temperature and the CORINE Land Cover in 1990; and (**c**) the area burned between 1990 and 2017.

### 2.2. CORINE Land Cover Changes and Wildfires

In the study of LULC, changes were used to form the maps of the CORINE Land Cover for 1990, 2000, and 2006 [58]. These maps were reclassified to SWAT land cover classes, which is the format read from ArcSWAT. The reclassification is found in the Supplementary Material (worksheet 1). Figures 2 and 3 illustrate the changes between the SWAT land cover classes of 1990 and 2000 and between the SWAT land cover classes of 1990 and 2006. These changes represented 6% of the basin area for the SWAT land cover classes of 1990 and 2000 (Figure 2a), and 10% of the basin area between 1990 and 2006 (Figure 2b). For better visualization and analysis, in Figure 3, only areas greater than 500 hectares are shown. The major changes in the area of SWAT land cover classes from 1990 to 2000 comprise the replacement of range—brush by forest (7080 ha), essentially coniferous (FRSE, 4009 ha) and deciduous forests (FRSD, 2212 ha), as well as burned areas (BARR, 1643 ha). The major change in the area of SWAT land cover classes from 1990 to 2006 was the increase in range—brush (RNGB, 14,384 ha) mainly from range—grasses (RNGE, 3635 ha), agricultural land—generic (AGRL, 2584 ha), forest—mixed (FRST, 2395 ha), forest—deciduous (FRSD, 1915 ha), forest—evergreen (FRSE, 1725 ha), and burned area (BARR, 1421 ha). As well as the replacement of agricultural land—row crops (AGRR) in agricultural land—generic (AGRL, 4155 ha).

The cartography of the burned areas covering the period 1990–2017 is available at the Conservation of Nature and Forests [61] (Table 1). In this period in the Sabor river basin, 103,033 ha was burned, which corresponds to 32% of the basin area (Figure 1c). The largest burned area (approximately

13,000 ha) occurred in 2013, followed by burned areas above 6700 ha, which occurred in 1994, 1998, 2000, 2012, and 2017 (Figure 4).

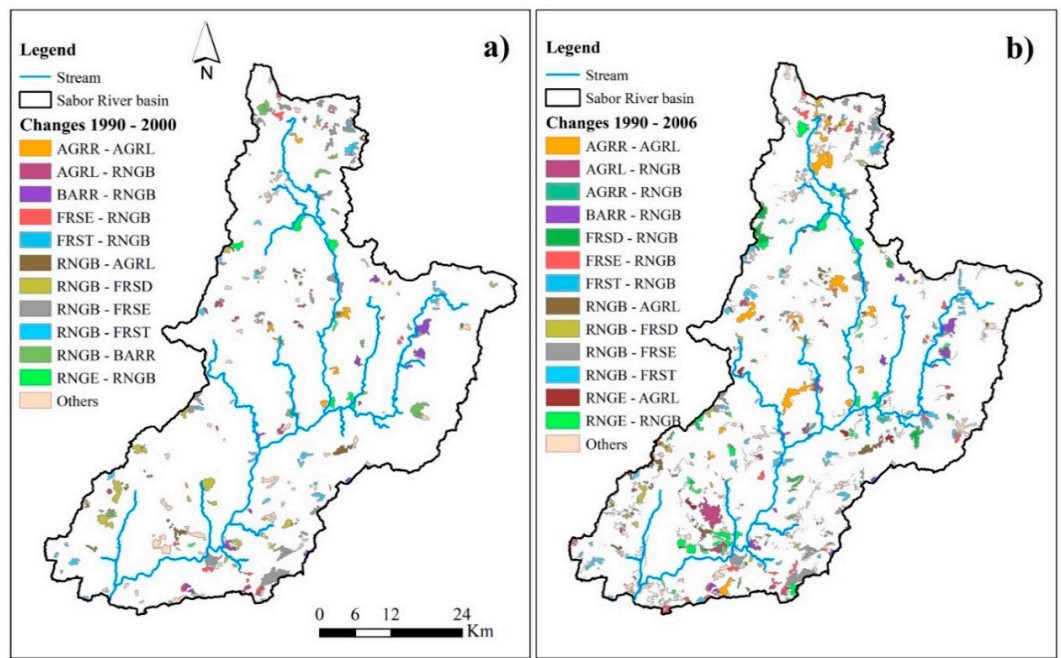

**Figure 2.** The spatial distribution of changes (**a**) between the CORINE Land Cover 1990 and 2000; and (**b**) between the CORINE Land Cover 1990 and 2006, in the Sabor river basin. Only the areas with changes greater than 500 hectares are presented. The SWAT codes are agricultural land—row crops (AGRR), agricultural land—generic (AGRL), range—brush (RNGB), range—grasses (RNGE), barren (BARR), forest—deciduous (FRSD), forest—evergreen (FRSE), and forest—mixed (FRST).

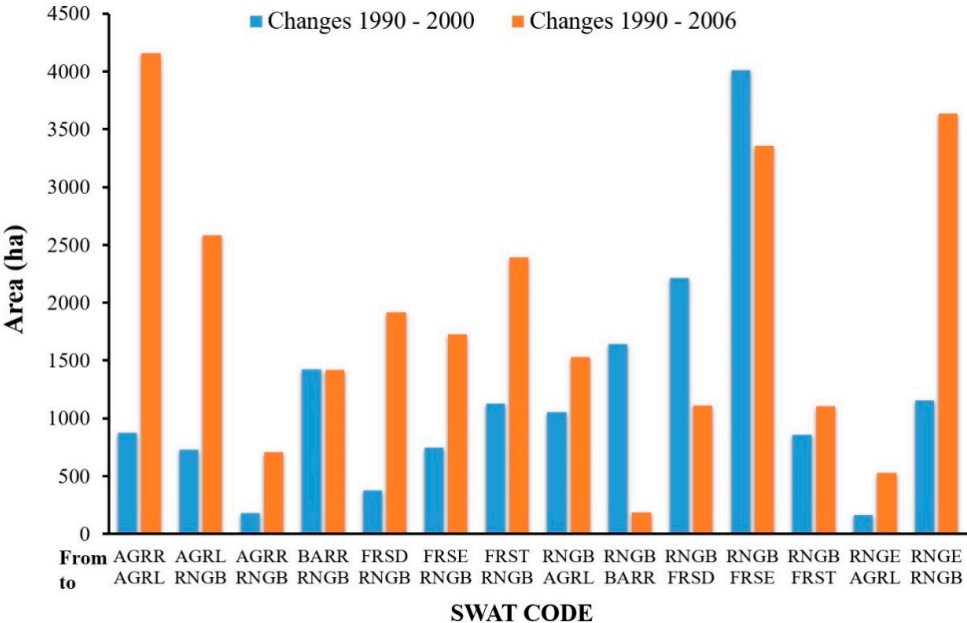

**Figure 3.** Graphic of changes between the CORINE Land Cover in 1990 and 2000, and between the CORINE Land Cover in 1990 and 2006 in the Sabor river Basin. Only the areas with changes greater than 500 hectares are presented. The SWAT codes are agricultural land—row crops (AGRR), agricultural land—generic (AGRL), range—brush (RNGB), range—grasses (RNGE), barren (BARR), forest—deciduous (FRSD), forest—evergreen (FRSE), and forest—mixed (FRST).

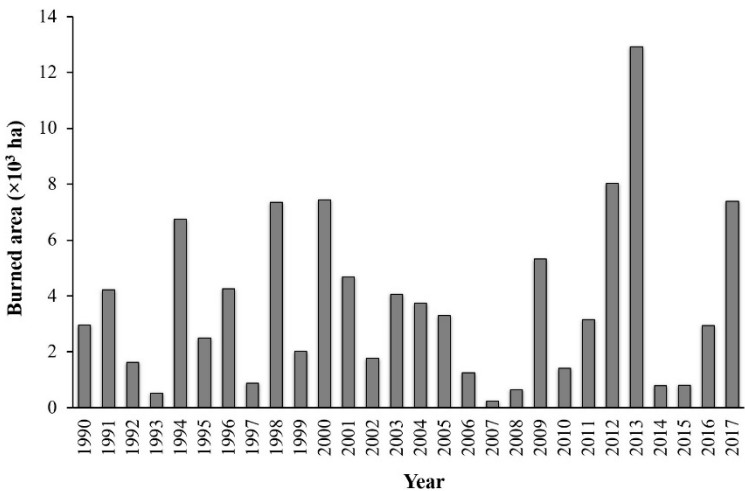

**Figure 4.** The area burned between 1990 and 2017 in the Sabor river basin.

**Table 1.** Maps and records were used in the construction of the hydrological model, in the evaluation of Land Use and Land Cover (LULC) changes between 1990 and 2008, as well as in the analysis of the afforestation and wildfire scenarios. All data types used have reference to the purpose, their owner institution, and the source of data.

| Data Type | Purpose | Owner Institution | URL of Internet Website |
|---|---|---|---|
| **Construction of the hydrological model** | | | |
| Digital Elevation Model | Calculation of the elevation and slope | Directorate-General of Territory | http://www.dgterritorio.pt/ |
| Drainage network | Definition of the stream network and delineation of the watershed | Portuguese Water Institute | http://geo.snirh.pt/AtlasAgua/ |
| Corine Land Cover 1990 | Creation of a land use grid for assessing vegetation cover type and calculation of curve number (CN) | European Environmental Agency | http://www.eea.europa.eu/ |
| Soils | Creation of a soil grid for assessment of soil types and calculation of CN | Directorate-General of Territory | http://scrif.igeo.pt/ |
| Records of weather stations | Construction of the hydrological model to estimate the water balance at the sub-basin level | Portuguese Water Institute | http://snirh.apambiente.pt/ |
| Records of hydrometric stations | Calibration of the hydrological model | Portuguese Water Institute | http://snirh.apambiente.pt/ |
| **Evaluation of the Land Use and Land Cover changes between 1990 and 2008** | | | |
| CORINE Land Cover in 2000 and 2006 | Assessment of the LULC changes between 1990 and 2008 | European Environmental Agency | http://www.eea.europa.eu/ |
| **Analysis of the afforestation and wildfires scenarios** | | | |
| Regional Forestry Management Plan of Douro and Nordeste to 2045 | Assessment of the impact of the afforestation projection on the water balance | Portuguese Institute for the Conservation of Nature and Forests | http://www.icnf.pt/ |
| Cartography of burned areas between 1990 and 2017 | Assessment of the impact of the wildfires scenario on the water balance | Portuguese Institute for the Conservation of Nature and Forests | http://www.icnf.pt/ |

*2.3. Conceptual Framework Model*

The conceptual framework model was developed to assess the effects of LULC changes and the forecast of afforestation and the occurrence of wildfires on water balance (Figure 5). SWAT was used to construct the hydrological model of the Sabor river basin with the following set of data: digital elevation model, land use, soil types, and weather station (Table 1). With these data, the drainage

network, the delimitation of the basin and sub-basins, and the hydrologic response units (HRU) were defined. After that, the model was calibrated by SWAT-CUP with streamflow discharge data on a daily basis. Then, the hydrological model was simulated with the values of the parameters obtained in the calibration procedure. This model was used in the diagnostic phase and in future scenarios. In the diagnostic phase, the model was used to assess the effects of LULC changes between 1960 and 2008 on water balance components with the CORINE Land Cover in 1990, 2000, and 2006. In future scenarios, the model was used to evaluate the effects of the forecast of afforestation and the occurrence of wildfires on water balance components.

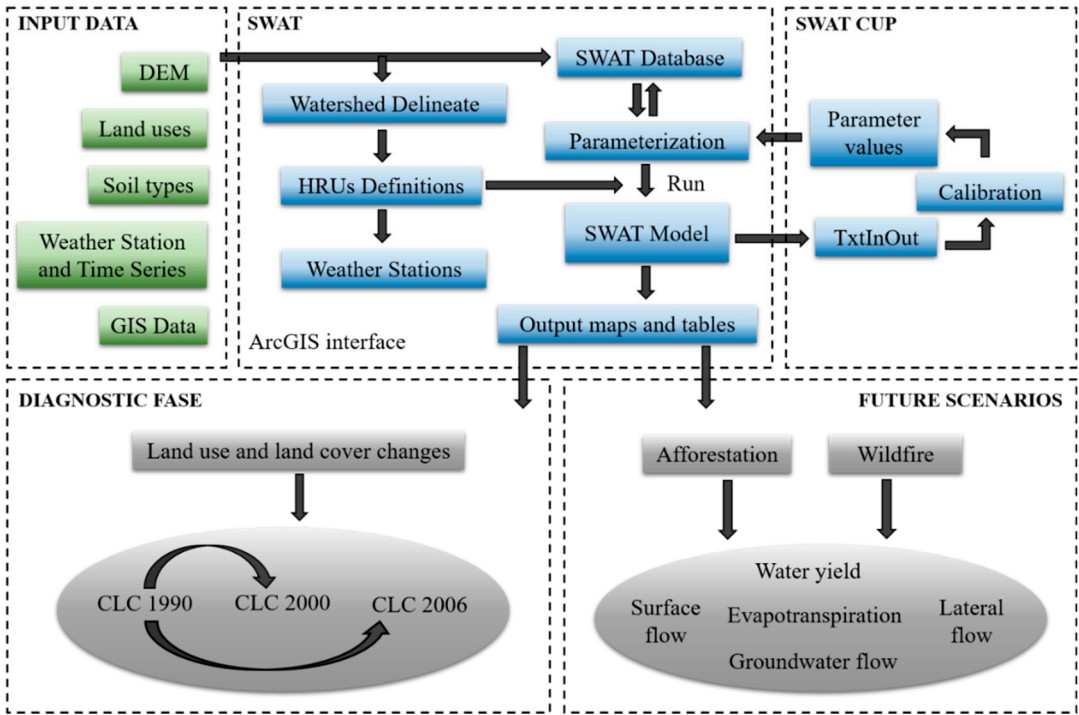

**Figure 5.** The conceptual framework model for constructing and modelling the water resources in the SWAT (Soil and Water Assessment Tool).

## 2.4. SWAT Model

The SWAT is based on a physical process to be simulated in a watershed [49]. In a watershed, SWAT continuously simulates the hydrological pattern processes, water balance, water quality, nutrients, and sediment exportation [49,50]. For simulations, information collected from various sources (e.g., topography, land cover) is necessary, and this is then assimilated into the database (e.g., soil characteristics, meteorological data). For modelling purposes, the SWAT model divided the watershed into a sub-basin linked in a cascade by the stream network [62]. In turn, the land area in a sub-basin is divided into HRUs. Each HRU is a portion of a sub-basin that comprises unique land cover, soil, and slope attributes [49]. The SWAT treats the HRU as a homogeneous unit of land use, management techniques, and soil properties and then quantifies the relative impacts of vegetation, management, soil, and climate change within each HRU [63]. The output of the hydrological model (e.g., runoff, sediments, nutrients) is calculated separately from each HRU and then added together to determine the total loading from a sub-basin. The advantage of HRUs is the increase in accuracy which adds to the prediction of loading from a sub-basin [49].

The SWAT uses a modified Soil Conservation Service Curve Number (SCS CN) methodology, and the Penman–Monteith equation to calculate the runoff and evapotranspiration, respectively [50]. Curve numbers have been developed and published for a wide range of land cover types and uses and can be found in [64]. The Penman–Monteith equation requires daily values of precipitation

(mm), maximum and minimum temperature (°C), solar radiation (MJ/m$^2$ day), relative humidity (%), and wind speed (ms$^{-1}$), but this observed data is usually available with gaps and thus may limit the performance and results of the model [49,62]. To fill the gaps, SWAT includes the WXGEN stochastic weather generator model to generate climatic data [63,65]. The WXGEN generates daily weather information that is missing from the monthly average data summarized over a number of years [49].

*2.5. SWAT Input Data*

The 2012 version of ArcSWAT was used to build a hydrological model of the Sabor river basin. ArcSWAT is an ArcGIS extension and graphic user input interface for SWAT, which is used worldwide and is continuously under development [66]. We selected ArcSWAT because it has been used in numerous hydrologic, decision-making, and environmental applications [62,65,67], and the authors have experience with using ArcGIS for the processing, overlay, and combination of multiscale and multi-type spatial data in thematic surveys or projects focused on the collection and interpretation of spatial data [34,68–73].

The input data used to construct the hydrological model were (i) the topography of Trás-os-Montes and Alto Douro to generate a digital elevation model and slope, (ii) the drainage network of the Sabor river basin to define the stream and delineate the basins and sub-basins, (iii) the CORINE Land Cover to create a land use grid, (iv) the soil types of Trás-os-Montes and Alto Douro to create a soil grid, and (v) weather data of stations located inside or near the basin to simulate climatic data (Figure 6, Table 1).

A precipitation dataset was compiled between 1957 and 2008 by the various climatologic stations located inside and near the basin (Figure 6b). The gaps in the daily precipitation time series were calculated by inverse distance weighted interpolation. This method is frequently used in climatic predictions and has already been proven to provide good results [17,62]. The time series of weather information insert in the WXGEN weather generator model was provided by the SNIRH meteorological station of Folgares (06N/01C) (Figure 6b). The WXGEN filled the daily missing data based on an average of 46 years of weather data.

A total of 37 sub-basins were defined in the basin area (Figure 1a), with an average area and standard deviation of 86 and 69 km$^2$, respectively. A total of about 523 HRUs were defined within sub-basins, with a threshold value of 10% for the land use, soil, and slope classes. The SWAT model was executed on a daily basis from 1957 to 1999 with a warm-up period of 3 years. A warm-up period is recommended to initialize the simulation process with the objective of ensuring the establishment of basic flow conditions and hydrologic processes equilibrium as well as to help to minimize the model values for the initial hydrological conditions [62,67].

The land use grid used in the construction of the hydrological model was the CORINE Land Cover 1990 (CLC) (Table 1), but this land cover does not provide information that corresponds to the SWAT land cover classes. Therefore, the CLC classes were reclassified, which led to several CLC classes having the same SWAT land cover classes (Figure 6c) [62,65]. For example, all classes of the heterogeneous agricultural areas of CLC classes were generalized into the SWAT land cover class agricultural land—generic. Among the different SWAT land cover classes, the most appropriate reclassification for the burned areas of CLC classes was barren (BARR). The reclassification is presented in the Supplementary Material (worksheet 1).

The State Soil and Geographic (STATSGO) is the soil database integrated into the SWAT [74]. These soil categories are unavailable in Portugal, and a match could not be found between those categories and the available ones for the research area. The soil map available in the Sabor river basin was extracted from the digital soil map of the Trás-os-Montes and Alto Douro region (Table 1). The soil categories were lithosols, cambisols, alisols, anthrosols, fluvisols, and urban land (Figure 6d). Therefore, data, including the soil component parameters and soil layer parameters, were inserted into the SWAT database for each soil type, except for urban land, which already existed.

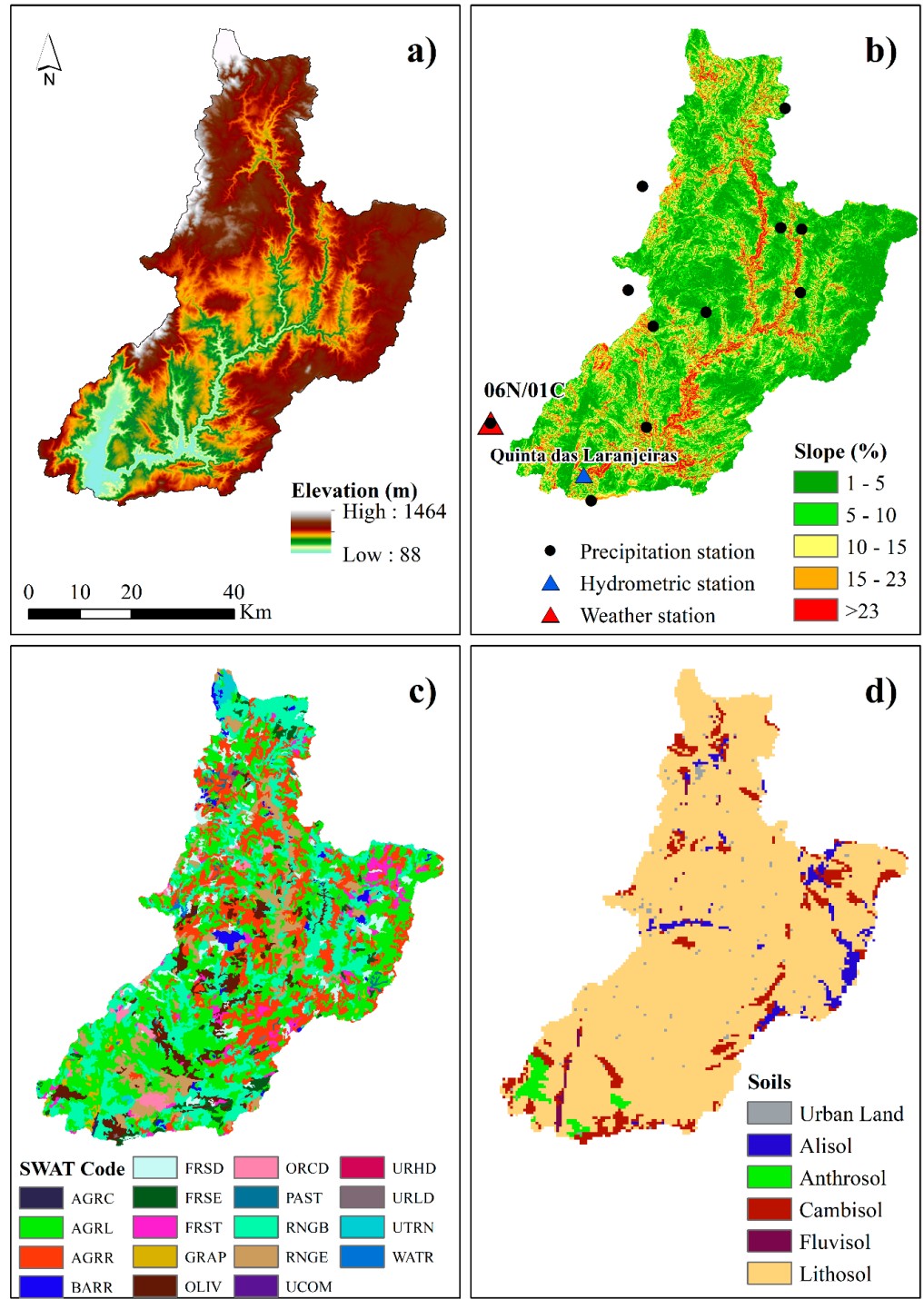

**Figure 6.** The SWAT (Soil and Water Assessment Tool) input data used for constructing the hydrological model were (**a**) the elevation; (**b**) the slope, meteorological, and hydrometrical data; (**c**) the SWAT codes of land cover classes; and (**d**) the soil types. The SWAT codes are agricultural land—close-grown (AGRC), agricultural land—generic (AGRL), agricultural land—row crops (AGRR), barren (BARR), forest—deciduous (FRSD), forest—evergreen (FRSE), forest—mixed (FRST), vineyard (GRAP), olives (OLIV), orchard (ORCD), pasture (PAST), range—brush (RNGB), range—grasses (RNGE), commercial (UCOM), residential—high density (URHD), residential—low density (URLD), transportation (UTRN), and water (WATR).

The hydrological model was calibrated from 1960 to 1999 and validated from 2000 to 2008. A hydrometric station called Quinta das Laranjeiras (Figure 6b) was used to calibrate and validate the streamflow on a daily basis.

## 2.6. SWAT-CUP Model Calibration

The computer program used for calibrating the SWAT 2012 models was the SWAT-CUP 2012 (Calibration and Uncertainty Procedures) [75]. The SWAT-CUP 2012 consists of five different calibration procedures, including functionalities for validation and sensitivity analysis. The calibration procedure SUFI-2 (Sequential Uncertainty Fitting) was used in this work. The SUFI-2 algorithm is quite efficient for large-scale, time-consuming models [75,76]. In SUFI-2, the uncertainty analysis is based on the discrepancy assessment between observed and simulated values, taking into account potential sources of uncertainty, like observed data, the conceptual model, and parameters. The degree of uncertainty is quantified by the 95% prediction uncertainty (95PPU), measured by the P-factor and R-factor. The P-factor is the percentage of measured data bracketed by the 95PPU and varying from 0 to 1, wherein 1 indicates a 100% bracketing fit of the observed values. The R-factor measures the calibration quality and indicates the thickness of the 95PPU. A P-factor of 1 and an R-factor of 0 mean a perfect fit for the observed and calibrated values [75].

SUFI-2 has several criteria and quantitative statistical methods to evaluate the outcome simulation values from SWAT, as compared with the observed data [75]. The objective function selected as the calibrated parameter set was the coefficient of determination ($R^2$). The statistical methods used to assess the model performance were the following: the Nash–Sutcliffe efficiency (NS), the percent bias (PBIAS), the ratio of the root mean square error to the standard deviation of measured data (RSR), and the coefficient of determination ($R^2$). The model performance is considered satisfactory whenever $R^2$ and NS are greater than 0.5, RSR is less than 0.7, and PBIAS is less than ±25% for streamflow [77,78].

## 2.7. Diagnostic Phase of Land Use and Land Cover Changes

In the diagnostic phase, CLC 1990, CLC 2000, and CLC 2006 were used to determine the effects of LULC changes on the water balance components of the Sabor river basin. The SWAT hydrological model was constructed and calibrated with CLC 1990. In forthcoming sections, this hydrological model will be referred to as the reference model. After that, two more hydrological models were constructed, one with the CLC 2000 and another one with the CLC 2006. The values of the calibration parameters of the reference model were inserted into these two hydrological models. Using the same parameter values in all hydrological models ensures that changes in the streamflow are exclusively due to LULC changes. Thus, the study of the LULC changes consisted of comparing the values of the water balance components between the reference model and both the CLC 2000 hydrological model and the CLC 2006 hydrological model. The reference model was simulated between 2000 and 2008, the CLC 2000 hydrological model was simulated between 2000 and 2005, and the CLC 2006 hydrological model was simulated between 2006 and 2008.

## 2.8. Future Scenarios of Afforestation and Wildfires

In the Sabor river basin, the afforestation scenario was based on the Regional Plan for Forest Management (RPFM) of Douro [79] and Northeast [80], and the wildfires scenario was based on the burned area between 1990 and 2007 [61]. The Regional Plan for Forest Management (Portuguese Regulate Decree no. 3/2007, published on 17 January 2007) includes a plan to be accomplished until 2045, and the main objectives are to reduce the risk of wildfire occurrence, and to adjust the proportions of resinous and deciduous species based on the application of correct forest management models. In the RPFM, the Sabor river basin occupies eleven homogeneous sub-regions in the Douro and Northeast regions, which are referenced in the Supplementary Material (worksheet 2).

In order to create a model of the afforestation scenario, a map was created with the coniferous and broad-leaved areas proposed by RPFM until 2045. Afforestation consisted of counting the areas of

coniferous, broad-leaved, and mixed forest in CLC 1990 and calculating the missing areas until reaching the percentage of afforestation proposed by the RPFM for each sub-region (Figure 7a). According to this technical report, the area of afforestation of agricultural land will include 40% coniferous forest and 60% broad-leaved forest, and the afforestation of semi-natural areas will include 70% coniferous forest and 30% broad-leaved forest [79,80]. To make the map, the afforestation of agricultural land was carried out in the following order of CLC classes: agro-forestry areas, land mainly occupied by agriculture, with significant areas of natural vegetation and complex cultivation patterns. The afforestation of semi-natural areas was done in the following order of CLC classes: transitional woodland-shrub, sclerophyllous vegetation, moors and heathland, and burned areas. A map of the afforested area is presented in Figure 7b, and calculation of the respective areas is found in the Supplementary Material (worksheet 2). The map of the afforestation scenario inserted in the SWAT was the CLC 1990 updated with afforestation areas. Also, the map of the wildfires scenario inserted in the SWAT was the CLC 1990 updated with all burned areas in the Sabor river basin between 1990 and 2017 (Figure 7d).

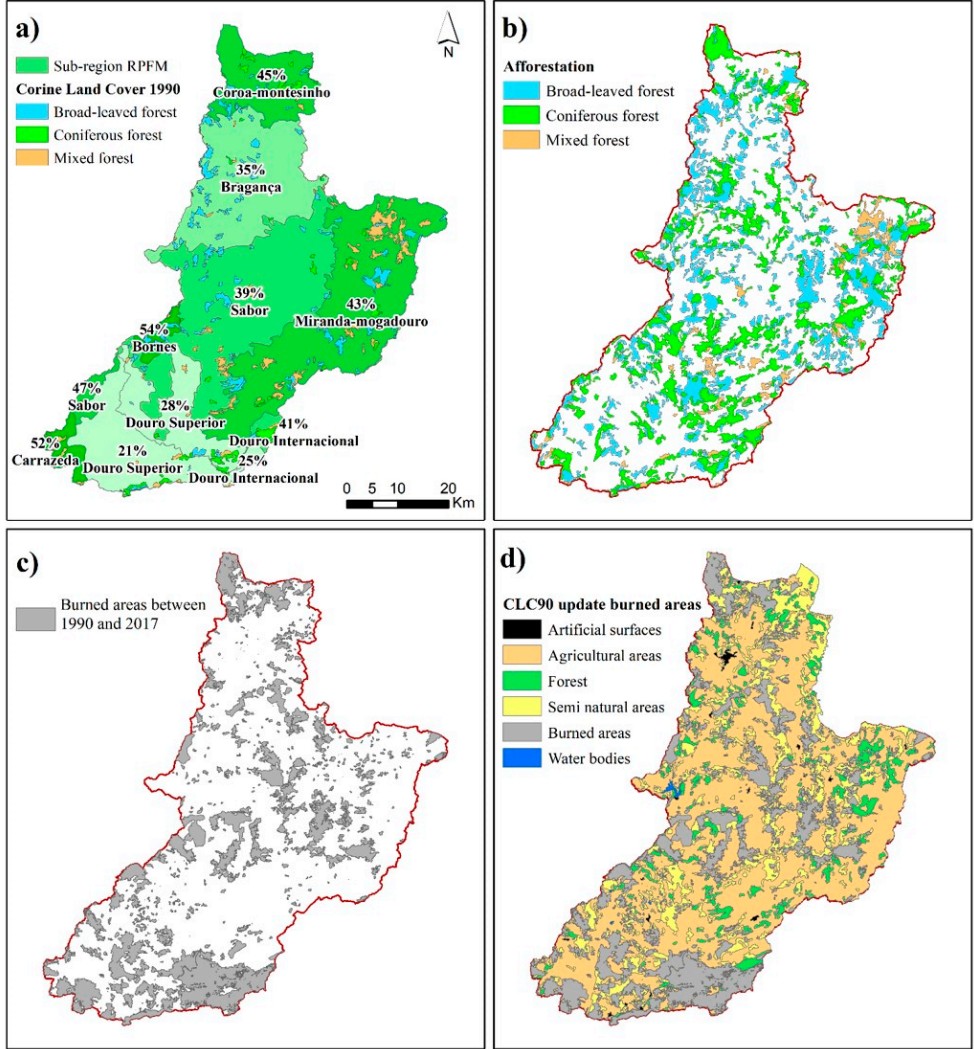

**Figure 7.** The maps of afforestation and wildfires scenarios inserted in the SWAT for constructing the hydrological models were (**a**) broad-leaved, coniferous, and mixed forest from the CORINE Land Cover 1990 and the projected percentages of afforestation per sub-region for 2045, proposed by the Regional Plan for Forest Management; (**b**) the afforestation projection for 2045; (**c**) areas burned between 1990 and 2017; and (**d**) the CORINE Land Cover 1990 updated with areas burned between 1990 and 2017.

After creating the maps, two hydrological models were constructed, one with the map of the afforestation scenario and another with the map of the wildfires scenario. The values of the calibration parameters of the reference model were inserted into these two hydrological models. Once again, the use of the same parameters in both hydrological models allowed the scenarios to be compared with the reference data. In other words, the factors used to compare the afforestation and wildfires scenarios with the reference model included the water yield surface flow, evapotranspiration, lateral flow, and groundwater flow.

## 3. Results

### 3.1. Calibration and Validation of the Streamflow

The streamflow was calibrated for a 39-year period (1960–1999) and validated for a 9-year period (2000–2008) both on a daily basis. The parameters and their respective values resulting from the calibration are shown in Table 2. The graphics in Figure 8a,b show good agreement between the observed and simulated streamflow for both calibration and validation. The calibration and validation, illustrated in Figure 8a,b, respectively, were performed on a daily basis, but for best visualization, they are represented on a monthly basis. The goodness-of-fit indicators for the streamflow calibration (Table 3), based on the $R^2$, RSR, and NS show satisfactory performances (with values of 0.63 and 0.62) and PBIAS shows a very good performance (2.7%) [50,51]. The same goodness-of-fit indicators were obtained for the validation with a very good performance for $R^2$ (with 0.8), and satisfactory performances for RSR, NS, and PBIAS with 0.63%, 0.61%, and −24%, respectively (Table 3). The negative value of PBIAS indicates a model overestimation bias [77].

**Table 2.** The parameters used in the calibration procedure of streamflow. In the legend of methods, R is relative and V is the replacement value.

| Method and Parameter | Description | Units | Minimum Value | Maximum Value | Fitted Value |
|---|---|---|---|---|---|
| R_CN2.mgt | Curve number for moisture condition II | – | −0.08 | 0.10 | 0.05 |
| V_ALPHA_BF.gw | Base flow alpha factor | days | 0.53 | 0.82 | 0.59 |
| V_GW_DELAY.gw | Flow delay time for aquifer recharge | days | −80.89 | 99.30 | 14.84 |
| V_GWQMN.gw | Flow threshold depth of water in the shallow aquifer | mm | 0.72 | 1.81 | 1.26 |
| V_REVAPMN.gw | Threshold depth of water in the shallow aquifer | mm | 256.09 | 383.16 | 295.00 |
| V_GW_REVAP.gw | Groundwater re-evaporation coefficient | – | 0.09 | 0.19 | 0.12 |
| V_RCHRG_DP.gw | Flow deep aquifer percolation coefficient | – | −0.05 | 0.69 | −0.03 |
| V_SHALLST.gw | Initial depth of water in the shallow aquifer | mm | −23476.17 | 9946.85 | −5302.40 |
| V_CH_N2.rte | Manning's "n" value in the main channel | – | 0.08 | 0.16 | 0.12 |
| V_CH_K2.rte | Effective hydraulic conductivity in the main channel | – | 56.47 | 213.88 | 147.97 |
| V_ALPHA_BNK.rte | Baseflow alpha factor for bank storage | – | 1.30 | 2.01 | 1.91 |
| V_ESCO.hru | Soil evaporation compensation factor | – | 0.53 | 0.90 | 0.88 |
| V_EPCO.hru | Plant uptake compensation factor | – | 0.08 | 0.58 | 0.39 |
| V_SLSUBBSN.hru | Average slope length | – | 171.37 | 248.23 | 214.12 |
| R_SOL_AWC (1).sol | Soil available water capacity (soil 1st layer) | – | 0.02 | 0.18 | 0.18 |
| R_SOL_K (1).sol | Saturated hydraulic conductivity | – | 301.52 | 633.36 | 411.44 |
| V_SURLAG.bsn | Surface runoff lag coefficient | – | 2.68 | 9.03 | 6.53 |
| V_CH_K1.sub | Effective hydraulic conductivity in the tributary channel alluvium | – | 108.63 | 194.72 | 178.04 |

**Table 3.** Goodness-of-fit indicators for daily calibration between 1960 and 1999 and validation of streamflow between 2000 and 2008 in the Sabor river basin.

| Measure | Calibration | Acceptable Ranges |
| --- | --- | --- |
| **Calibration** | | |
| $R^2$ (coefficient of determination) | 0.63 | > 0.5 acceptable [51] |
| RSR (standardized RMSE) | 0.62 | Satisfactory [50] |
| NS (Nash–Sutcliffe coefficient) | 0.62 | Satisfactory [50] |
| PBIAS (percent bias) | 2.7% | Very good [50] |
| **Validation** | | |
| $R^2$ (coefficient of determination) | 0.80 | > 0.75 very good [51] |
| RSR (standardized RMSE) | 0.63 | Satisfactory [50] |
| NS (Nash-Sutcliffe coefficient) | 0.61 | Satisfactory [50] |
| PBIAS (percent bias) | −24% | Satisfactory [50] |

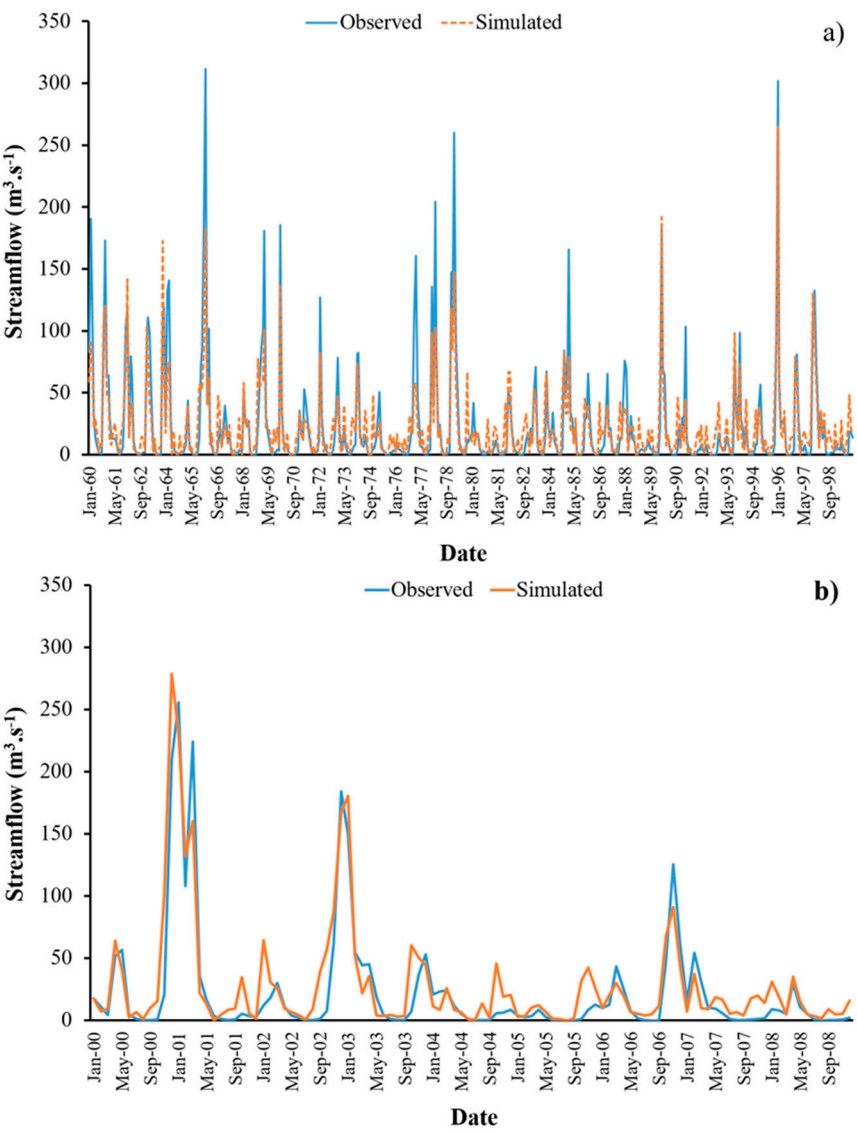

**Figure 8.** The comparison of observed and simulated streamflow during (**a**) the calibration (between 1960 and 1999); and (**b**) validation (between 2000 and 2008) in the Sabor river basin. The simulation of the streamflow was executed on a daily basis, but for best visualization was present on a monthly basis.

### 3.2. Land Cover and Land Use Changes

Figure 9 shows the graphics of the water balance components of the reference model (with the CLC 1990, simulated between 2000 and 2008) the hydrological model of the CLC 2000 (simulated between 2000 and 2005), and the hydrological model of the CLC 2006 (simulated between 2006 and 2008). The results show that the LULC changes that occurred in 2000 and 2006 led to a decrease in the water yield and an increase in evapotranspiration (Figure 9a,b). The water yield decreased by an average of 91 and 52 mm·year$^{-1}$ for the LULC changes in 2000 and 2006, respectively. The evapotranspiration increased by an average of 90 and 55 mm·year$^{-1}$ for the LULC changes in 2000 and 2006, respectively. The values of surface flow and groundwater flow decreased, while the lateral flow increased (Figure 9c,e). On average, for the LULC changes in 2000 and 2006, the surface flow decreased by 28 and 23 mm·year$^{-1}$, and the groundwater flow decreased by 91 and 50 mm·year$^{-1}$, respectively. The lateral flow increased by 10 and 5 mm·year$^{-1}$ for the LULC changes in 2000 and 2006, respectively.

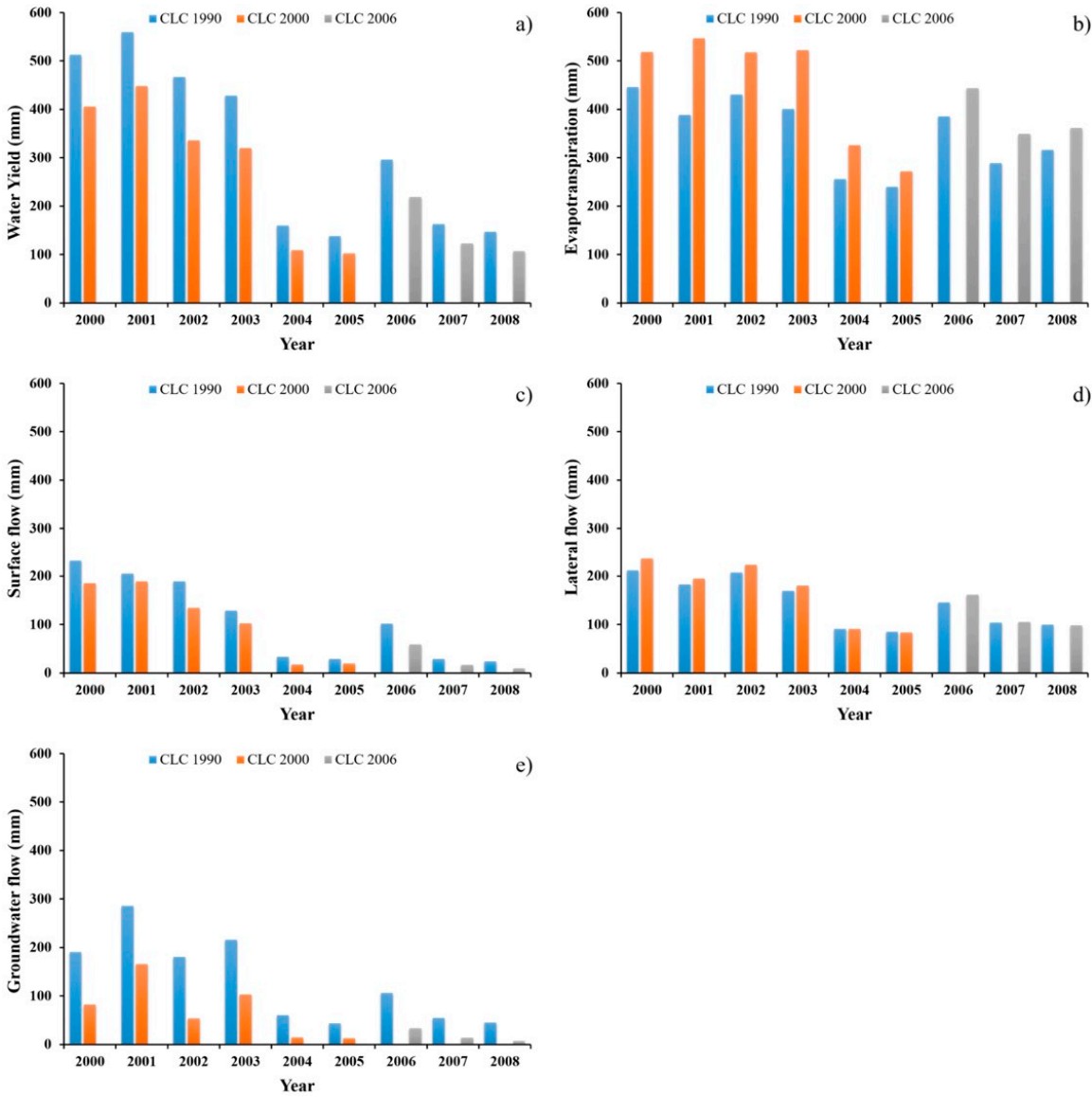

**Figure 9.** Quantity of water (mm) (**a**) in the water yield; (**b**) evapotranspiration; (**c**) surface flow; (**d**) lateral flow; and (**e**) groundwater flow. The blue bars are the reference model (simulated between 2000 and 2008 with the CLC 1990), the orange bars are the CLC 2000 model (simulated between 2000 and 2005), and the grey bars are the CLC 2006 model (simulated between 2006 and 2008).

*3.3. Afforestation and Wildfires Scenarios*

The values of the water balance components were calculated on a monthly basis between the reference model (CLC90) and both the afforestation and wildfires scenarios. The results are illustrated in Figures 10 and 11, which represent the monthly values and spatial distribution, respectively. The graphics of Figure 10a,b show that, in both scenarios and in every month of the year, the water yield decreased and the evapotranspiration increased. The decrease in the water yield was more pronounced in the rainy season (autumn and winter) than in the dry season (spring and summer) and in the afforestation scenario compared with the wildfires scenario. In the afforestation scenario, these large decreases varied from 28 hm$^3$·year$^{-1}$ in October to 62 hm$^3$·year$^{-1}$ in January, while in the wildfires scenario they were somewhat smaller, varying from 15 hm$^3$·year$^{-1}$ in October to 49 hm$^3$·year$^{-1}$ in January. The month of August registered much lower water yield decreases: 3 hm$^3$·year$^{-1}$ in the afforestation scenario and 2 hm$^3$·year$^{-1}$ in the wildfires scenario. The evapotranspiration increase was more pronounced in January, February, and March, with values ranging from 46 to 82 hm$^3$·year$^{-1}$ for the afforestation scenario and from 33 to 61 hm$^3$·year$^{-1}$ for the wildfires scenario. In the afforestation scenario, the lowest increase in evapotranspiration was registered in August (3 hm$^3$·year$^{-1}$), while in the wildfires scenario, a small decrease in evapotranspiration was registered in September (0.4 hm$^3$·year$^{-1}$).

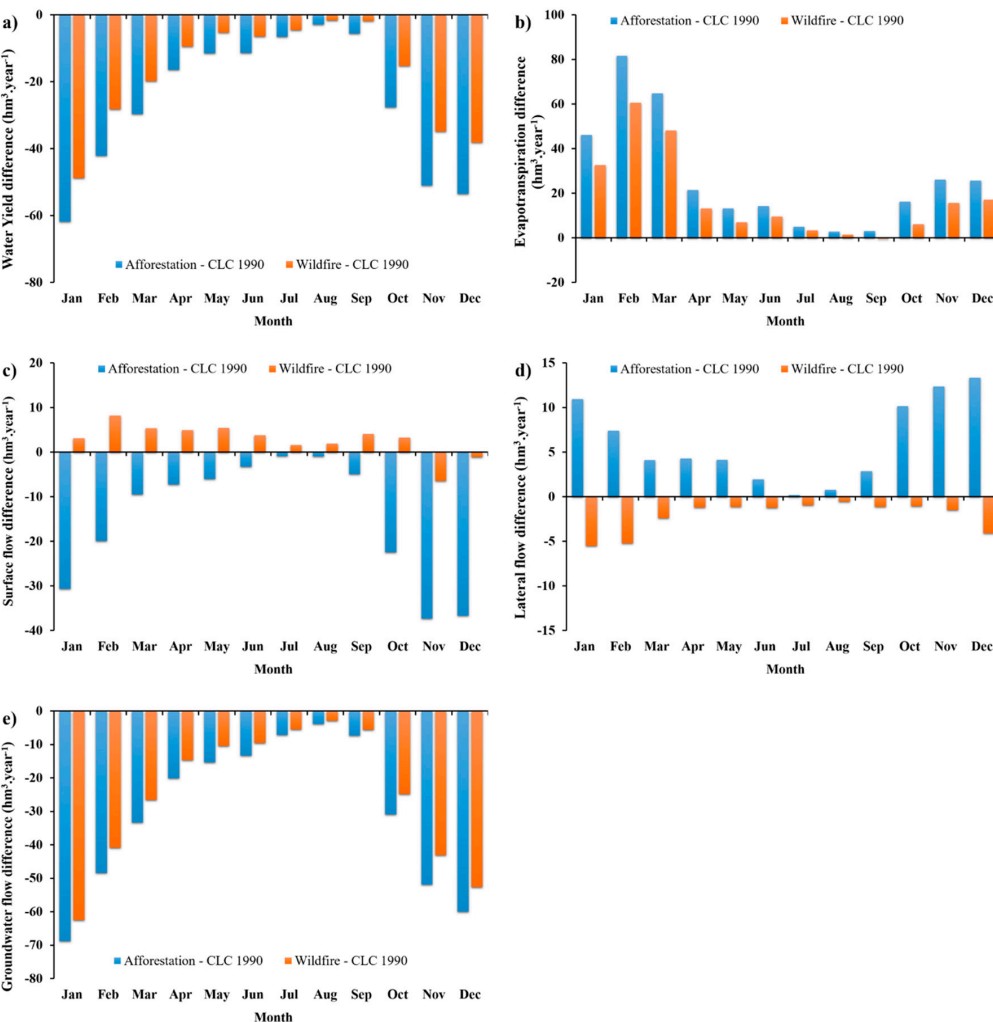

**Figure 10.** Quantity of water (hm$^3$·year$^{-1}$) (**a**) in the water yield; (**b**) in evapotranspiration; (**c**) in surface flow; (**d**) in lateral flow; and (**e**) in groundwater flow. The blue bars show the difference between the reference model and the afforestation scenario model, and the orange bars show the difference between the reference model and the wildfires scenario model.

The water available for surface flow and lateral flow showed different behaviors in both scenarios. In the afforestation scenario, the surface flow decreased and the lateral flow increased, and the opposite occurred for the wildfires scenario (Figure 10c,d). In the afforestation scenario, the major decrease in the surface flow occurred in January, November, and December, with values ranging from 31 to 37 $hm^3 \cdot year^{-1}$, while the smallest decrease occurred in July with 0.8 $hm^3 \cdot year^{-1}$. The major increase in the lateral flow occurred between October and January with values ranging from 10 $hm^3 \cdot year^{-1}$ in October to 13 $hm^3 \cdot year^{-1}$ in December, while the smallest increase occurred in July (0.2 $hm^3 \cdot year^{-1}$). In the wildfires scenario, the surface flow increased between January and October with values ranging between 2 $hm^3 \cdot year^{-1}$ in July and 8 $hm^3 \cdot year^{-1}$ in February, and it decreased in November and December with values of 1 and 6 $hm^3 \cdot year^{-1}$, respectively. The lateral flow decreased in every month of the year, with values ranging from 1 $hm^3 \cdot year^{-1}$ in August to 6 $hm^3 \cdot year^{-1}$ in January. The major decrease in the groundwater flow occurred in the rainy season in both scenarios (Figure 10e). In the afforestation scenario, these large decreases varied from 31 $hm^3 \cdot year^{-1}$ in October to 69 $hm^3 \cdot year^{-1}$ in January, while in the wildfires scenario, they were somewhat smaller, varying from 25 $hm^3 \cdot year^{-1}$ in October to 62 $hm^3 \cdot year^{-1}$ in January. The lowest groundwater flow was registered in August, with 4 $hm^3 \cdot year^{-1}$ in the afforestation scenario and 3 $hm^3 \cdot year^{-1}$ in the wildfires scenario.

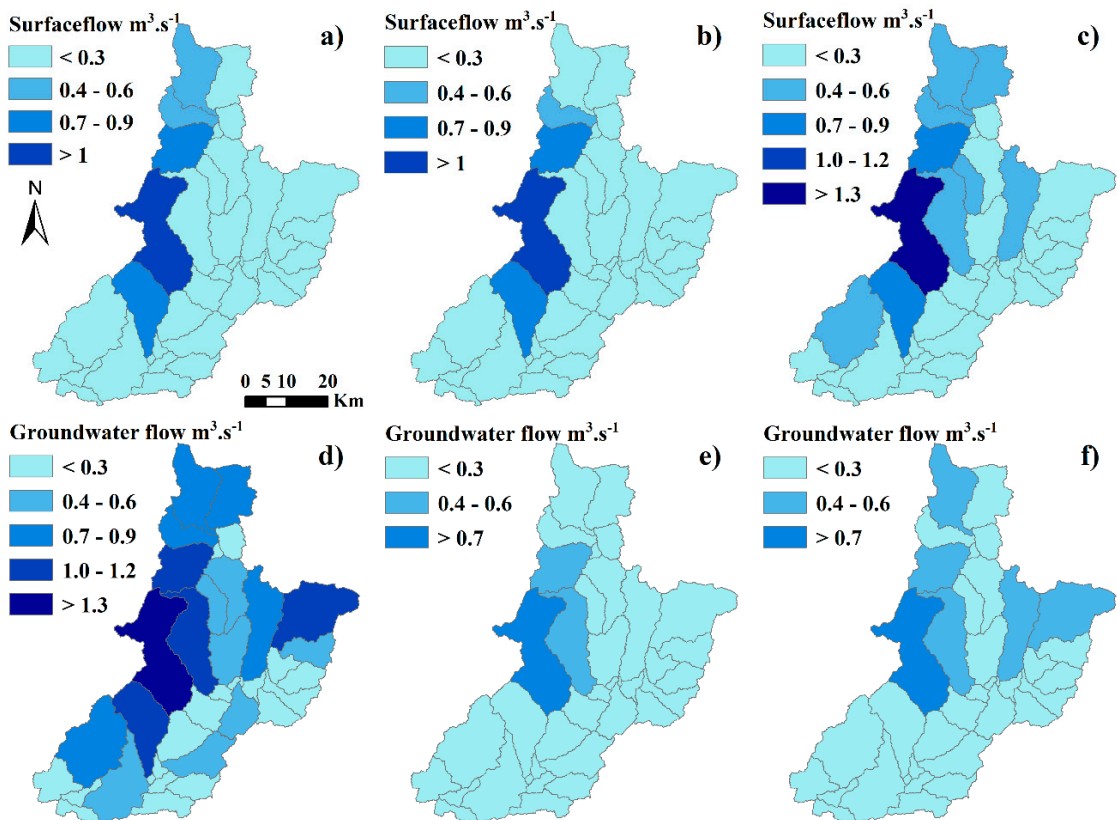

**Figure 11.** SWAT output maps of (**a**–**c**) the surface flow and (**d**–**f**) the groundwater flow. The surface flow matches (**a**–**c**) the reference model, the afforestation scenario, and the wildfires scenario, respectively. The groundwater flow matches (**d**–**f**) the reference model, the afforestation scenario, and the wildfires scenario, respectively.

SWAT output maps of the surface flow and the groundwater flow for the reference model and scenarios are illustrated in Figure 11. Figure 11a–c shows the surface flow of the reference model, for the afforestation scenario, and for the wildfires scenarios, respectively. Figure 11d–f shows the groundwater flow of the reference model, for the afforestation scenario, and for the wildfires scenario, respectively. The maps show that the highest surface flow and groundwater flow values in the

reference model (Figure 11a,d) and scenarios (Figure 11b,c,e,f) are located in the sub-basins of the Northwest. When comparing the maps of the reference model with the scenarios, it can be seen that the surface flow and groundwater flow values decrease. However, these decreases are major in the afforestation scenario.

## 4. Discussion

### 4.1. Performance of Streamflow and TP Calibration

The performance statistics of the model for the daily streamflow during calibration and validation periods were satisfactory and good, respectively. Similar results were achieved by other authors [81,82] using the SWAT for streamflow simulation in both the calibration and validation periods in the Alto Sabor river basin (located in the upstream part of the Sabor river basin). The results of the performance statistics indicated that SWAT was able to capture and reproduce the average flows and seasonal variations of the Sabor river basin (Figure 8a). However, the SWAT also shown had difficulty reproducing the observed extreme values of the streamflow, as shown by the regular underestimation of the most important peak events for the calibration period. Many studies have also detected the inability of the SWAT models to reproduce the highly complex hydrological processes during the high flow periods [52,53,81,82]. Major high flows or extreme conditions occurred in the winter of the years 1960, 1966, 1969, 1985, and between 1976 and 1979 (Figure 8a). Several authors analyzed how the rainfall-runoff translation occurs in reality compared with that simulated by the model [52]. These authors verified that the same amount of annual rainfall did not correspond to the same amount of observed streamflow. In contrast, the model presented a similar streamflow for the same amount of rainfall. The authors stated that the curve number method in SWAT uses the average daily rainfall and does not take into account the intensity and duration of rainfall. This is a factor that can explain why peak flows were underestimated by the model. One example of the high intensity of rainfall was registered in four basins of the North of Portugal, between 2000 and 2006, which had a considerable number of days that exceeded 20 mm of precipitation [68].

Despite the goodness-of-fit indicators for the streamflow calibration being considered satisfactory, there are some potential factors which compromise the performance of the model. We identified the following potential factors: (i) the climate stations cannot represent the orographic effect on the precipitation rates, (ii) the meteorological data are considerably affected by orographic variations, (iii) the human activities across the catchment may not have been accounted for in the model (e.g., artesian wells and boreholes, irrigation, and domestic consumption), (iv) the observed flow values are not 100% accurate. The precipitation gauges are located between 441 and 905 m of elevation with an average altitude of 659 m, and they cannot represent the orographic effect which varies between 88 and 1464 m of elevation. The orographic variations have a considerable effect on meteorological data; for example, the vertical temperature gradient, the depressions, and the shade in the valley result in lower minimum temperatures. These local variations are not entirely covered by weather stations. The last factor that influences the effectiveness of calibration is the problem that the hydrometric gauges faced during data collection. The discharge flow was estimated from hydrometric levels which were transformed into streamflow by a state discharge curve, but it was only valid within a range of hydrometric heights. Thus, the calculation used was not able to capture the actual hydrological pattern within extreme events.

### 4.2. Effects of LULC Changes and Future Scenarios for Water Resources

A major change in the LULC was the replacement of range—brush (scrub and herbaceous vegetation associations) with forest (essentially coniferous and deciduous) from CLC 1990 to CLC 2000 and the replacement of range—grasses (natural grasslands), agricultural land—generic (heterogeneous agricultural areas), forest (coniferous, deciduous, and mixed) and burned area in range—brush from CLC 1990 to CLC 2006 (Figure 3). These LULC changes are a result of a complex process of

plant recolonization of shrubs and forests as a consequence of farmland abandonment. This natural reforestation has the advantage of limiting soil erosion. However, the authors of [56,83] showed that many Mediterranean areas evolved into simplified and continuous landscapes with plenty of flammable wood, which led to fire occurrences. The wildfires increased soil erosion and progressively reduced the potential for recolonization of plants, leading to stony soils. This explains the many degraded panoramic landscapes in Mediterranean areas [16].

The results showed that the LULC changed from CLC 1990 to 2000, and from CLC 1990 to CLC 2006, causing decreases in the water yield, surface flow, and groundwater flow and increases in evapotranspiration and lateral flow. The major changes were observed in groundwater flow and evapotranspiration. The decrease in the groundwater flow in 200 and 2006 was, on average, 91 and 50 mm·year$^{-1}$, respectively (Figure 9a). The increase in evapotranspiration in 2000 and 2006 was, on average, 90 and 55 mm·year$^{-1}$, respectively (Figure 9b). The same results were obtained with the afforestation projection scenario proposed by the Regional Plan for Forest Management to 2045. In both the afforestation scenario and LULC changes, the decreases in the water yield, surface flow, and groundwater flow were due to a large amount of water being lost by evapotranspiration.

The increase in the lateral flow suggests that the forest provides an increase of water infiltration into the soil, but the high transpiration rate of the trees removes the water from the groundwater into the atmosphere. The increase in soil water infiltration by afforestation was confirmed by Ilstedt et al. [12]. The authors applied a meta-analysis based on several articles and concluded that the infiltration capacity increased, on average, by approximately three-fold after afforestation in agricultural fields. Another study [11] selected 20 studies to compare water flows in tropical watersheds under natural or planted forests and non-forest lands to provide useful results for valuing the watershed ecosystem services. The planted forests were Pines and Eucalyptus and lowland natural forests. The main results showed significantly lower total flows or water yields and groundwater flows under planted forests than under non-forest land uses. These results were explained by the high transpiration rates of Eucalyptus. According to these results, the authors argue that the effect of natural forests on base flow results from two competing processes: the high infiltration under the forest contributes to soil water recharge and the base flow increase, while the high transpiration of the trees contributes to the base flow decrease. Furthermore, the infiltration of soil water may be higher under planted forests than under non-forest land uses, but may not be sufficient to offset the loss of water by transpiration [12]. The compilation of paired-watershed results, based on a total of 137 basins, showed that reforestation results in a decrease in water yield and deforestation results in an increase [3]. Several authors have observed that reforestation could reduce the amount of sediment entering streams [84], as vegetation reduces the streamflow and increases infiltration [8,12]. The results of Serrano-Muela et al. [85] confirmed that forest conservation in the Central Spanish Pyrenees reduces floods and soil erosion, particularly on steep slopes. In this study, the major decreases in water yield, surface flow and groundwater flow, as well as the increases in lateral flow and evapotranspiration, were obtained in the wet season. For example, the decrease in water yield varied between 28 hm$^3$·year$^{-1}$ in October and 62 hm$^3$·year$^{-1}$ in January. Moreover, an increase in the lateral flow occurred between October and January with values ranging from 10 hm$^3$·year$^{-1}$ in October to 13 hm$^3$·year$^{-1}$ in December. The evapotranspiration was also higher in winter, because there was more water available to evaporate, but the highest values were obtained in February and March (with values ranging between 46 and 82 hm$^3$·year$^{-1}$). The reason for this is the combination of the rainfall and the increase of temperature that support evaporation and the growth of the canopy trees, which increases transpiration.

In contrast, in the wildfires scenario, increases in the surface flow and evapotranspiration and decreases in the lateral flow and groundwater flow were observed (Figure 10b–e). Similarly to the afforestation scenario, the major increases or decreases of water in the wildfires scenario mainly occurred in winter. For example, the water yield varied between 15 hm$^3$·year$^{-1}$ in October and 49 hm$^3$·year$^{-1}$ in January, and the increase in evapotranspiration was more pronounced in January, February, and March, with values ranging between 33 and 61 hm$^3$·year$^{-1}$.

Wildfires are agents of change that can dramatically alter evaporation and transpiration rates, and their short-term effects on surface evaporation rates may be complex. The increase in runoff after the fire has been attributed to destroyed vegetation as well as to reduced evapotranspiration [16,86–88]. The study by Miranda et al. [89] showed that immediately after a fire, evapotranspiration rates decrease. However, this situation is reversed with the first rains, because the burned area rapidly becomes a stronger sink for $CO_2$ and has higher evapotranspiration rates than nearby unburned areas. This difference persists throughout the wet season and is attributable to the greater physiological activity of the growing vegetation in the burned area. Similar rates of evaporation immediately after rainfall for both burned and unburned plots were observed in Campo Sujo, near Brazil [90]. However, within a few days of the soil drying, lower evaporation rates from the burned plot were observed. About 1 month after the fire, evaporation rates from the burned plot were typically as much as 80% that of the unburned control early in the wet season [90]. This was attributed to greatly enhanced rates of soil evaporation in the burned plot, especially immediately after the rain. Nagra et al. [91] also obtained similar results and justified the increase in the evaporation rates post-fire as a result of low albedo and reduced vegetation cover (lack of shading).

The increase in surface flow and the decreases in lateral flow and groundwater flow were due to the reduced infiltration capacity of the soil caused by the removal of vegetation by fire. Previous studies [19,92,93] showed that changes in vegetation cover and topsoil (composed of organic matter) have important impacts on the hydrological regime. A review of the literature developed in the European Mediterranean showed that wildfires make the soil more susceptible to removal by water erosion, less likely to allow infiltration, and more likely to promote surface flow [16]. The increase in surface flow, as well as soil erosion caused by fires, were proven in several studies in Portugal [19,92,93] and in the Mediterranean region [16,94,95]. For example, in burnt Eucalyptus and Pinus forest plantations in the Águeda Basin, North-Central Portugal, the surface flow on burned plots was 5%–25% higher than on unburnt ones [96]. For burned pine plantations in a small catchment located in Central Portugal, a runoff of up to 48.5% was found (1.1 km$^2$) [19]. In the Arbúcies basin, North-East Spain, there was an increase of 30% in flood runoff [94]. In the Rimbaud catchment in South-East France, an increase in the annual runoff of 30% was also verified during the first year after the fire [86].

A reduction in the infiltration capacity and the absence of vegetation improved the flood peak and soil erosion [16,93]. Nunes et al. [1] reported that during the study period (2005 and 2006), more than 80% of the total rainfall fell in autumn and winter. Consequently, the monthly rainfall erosivity, based on the Modified Fournier Index, increased considerably during these months. This result suggests that the energy available for erosion and transport was highly concentrated in this season. They also verified that the significantly larger amounts of runoff caused high soil loss. The overland flow was three times higher in 2006 than in 2005 which resulted in a sediment yield twice as high. However, the soil losses fundamentally depended on the amount and intensity of rainfall. The stability of the soil structure and aggregates is usually thought to be reduced by fire, producing more easily eroded soil [16].

Even so, modest post-fire soil losses could be important for soil longevity in some areas, because of soil organic matter and nutrient losses in solution or adsorbed onto eroded sediment particles [16,97]. Much of the post-fire nutrient content is in the form of ash, which is prone to removal by wind and water [19]. However, it can also benefit the soil quality through nutrient-rich ash becoming incorporated into the soil [16]. The high fire frequency and thin soils in a nutrient-poor ecosystem, typical of many fire-prone Mediterranean areas causes the risk of soil fertility depletion to be high [16,19].

### 4.3. Overview of the Effects of the LULC Changes on Water Resources

Changes in the LULC of the Sabor river basin have had impacts on the water balance. The main changes were the increases in forest and shrubs between 2000 and 2008. These changes caused decreases in the water yield and groundwater flow and an increase in evapotranspiration as well as the afforestation scenario. The decrease in water yield was due to the reductions of surface flow and

groundwater flow. The decrease in the surface flow was due to the increase in soil water infiltration by forests. Additionally, the decrease in groundwater flow can be partly attributed to the high transpiration rate of the trees. Despite forests causing substantial reductions in water yield, they also improve the water quality, reduce erosion, and benefit biodiversity. According to Francis et al. [10], a low to intermediate tree cover can improve soil hydraulic properties by up to 25 m from its canopy edge, which means that the hydrologic gains can be proportionally higher than the additional losses from the increased transpiration.

In the afforestation and wildfires scenarios, a decrease in the water yield was observed as well as an increase in evapotranspiration. The increase in evapotranspiration in the wildfires scenario was due to the evaporation of the water as a result of low albedo and a reduction in the vegetation cover. The comparison of the two scenarios showed that the trees were responsible for the reduction of groundwater flow. However, they contributed to the decrease in sediment yield and nutrients in surface flow. On the other hand, the wildfires scenario increased the surface flow as well as the sediment yield and the concentration of nutrients in the surface flow. The recurrence of fires has long-term implications because of the increase in soil erosion and the loss of nutrients, making it difficult for plants to recolonize. The water quality was affected by the increase in nutrient concentration due to eutrophication of the aquatic environment when associated with the increase in temperatures in the summer. The construction of two dams in the Sabor river for the production of electricity has increased the problem of eutrophication. According to the technical report [98] and the article [99], the changes in water quality caused by dam construction and the consequent stream water impoundment were significant. The increases in temperature and electric conductivity and accumulation of phosphorus and nitrogen in the reservoirs triggered the growth of algae, the increase of chlorophyll a, and a drop in the transparency of the water. The consequences of water deterioration for the aquatic fauna were severe, marked by abrupt declines of native fish species and the invasion of exotic species.

## 5. Conclusions

In this study, the SWAT2012 model was applied to assess the spatiotemporal dynamics of the water balance in the Sabor river basin. First of all, the LULC changes between 1960 and 2008 were analyzed with the CLC 1990, CLC 2000, and CLC 2006. Secondly, two scenarios were created, afforestation and wildfires. The afforestation scenario used was the projection proposed by the Regional Plan for Forest Management until 2045. The wildfires scenario used was based on areas burned in the Sabor river basin between 1990 and 2017.

The overall results of LULC changes between 1960 and 2008 show that the increase in forest and shrubs is a consequence of farmland abandonment in a basin where the population density is less than 20 inhabitants per $km^2$. The changes caused decreases in the water yield, surface flow, and groundwater flow and increases in evapotranspiration and lateral flow. The major changes occurred in the water yield and evapotranspiration. The water yield decreased in 2000 and 2006, on average, by 91 and 52 mm·year$^{-1}$, respectively. Evapotranspiration increased in 2000 and 2006, on average, by 90 and 55 mm·year$^{-1}$, respectively. The decrease in groundwater flow and increase in evapotranspiration were attributed to the high transpiration rate of the trees. Similar results were obtained for the afforestation scenario: decreases in the water yield and groundwater flow and an increase in evapotranspiration. The major decrease occurred in winter for groundwater flow with values varying between 28 hm$^3$·year$^{-1}$ in October and 62 hm$^3$·year$^{-1}$ in January. The increase in evapotranspiration was more pronounced in January, February, and March, with values ranging from 46 to 82 hm$^3$·year$^{-1}$.

In contrast, in the wildfires scenario, increases in the surface flow and evapotranspiration and decreases in the lateral flow and groundwater flow were observed. Similarly to the afforestation scenario, the major increases or decreases of water in the wildfires scenario occurred in winter. The major decrease in groundwater flow occurred in winter, for which values varied between 25 hm$^3$·year$^{-1}$ in

October and 62 hm³·year⁻¹ in January. The increase in evapotranspiration was more pronounced in January, February, and March, with values ranging from 33 to 61 hm³·year⁻¹.

The afforestation projection, proposed by the Regional Plan for Forest Management, will cause greater decreases in water yield, surface flow and groundwater than the wildfires scenario. A decrease in groundwater has implications for streamflow, because it is this flow that feeds the rivers in the dry season. However, it generates an increase in lateral flow, with more available water for irrigation. This is an important factor, because the basin is essentially agroforestry. With this study, we have shown that afforestation cannot simultaneously maximize all environmental benefits, but any form of afforestation should provide environmental improvements over agricultural land. Reductions in water yields could be minimized by planting trees with low water use at low densities and by avoiding landscape positions with access to the groundwater. An improvement in water quality could be achieved by using the contrasting strategies.

**Supplementary Materials:** The following are available online at http://www.mdpi.com/2073-4441/11/7/1464/s1, The Supplementary Materials comprise the following worksheets: Worksheet 1—Reclassification of CORINE Land Cover classes into SWAT counterparts; Worksheet 2—Forest occupation and predicted afforestation until 2045, in the eleven homogeneous sub-regions in the Douro and Northeast regions.

**Author Contributions:** Conceptualization, R.M.B.S. and F.A.L.P.; methodology, R.M.B.S. and F.A.L.P.; software, R.M.B.S.; validation, F.A.L.P.; formal analysis, L.F.S.F.; investigation, R.M.B.S.; resources, L.F.S.F. and R.M.V.C.; data curation, R.M.B.S., F.A.L.P., L.F.S.F., and R.M.V.C.; writing—original draft preparation, R.M.B.S.; writing—review and editing, F.A.L.P.; visualization, R.M.B.S.; supervision, F.A.L.P. and L.F.S.F.; project administration, R.M.V.C.; funding acquisition, R.M.V.C.

**Funding:** This research was funded by the INTERACT project "Integrated Research in Environment, Agro-Chain and Technology", no. NORTE-01-0145-FEDER-000017, in the line of research entitled BEST "Bio-economy and Sustainability", and co-financed by the European Regional Development Fund (ERDF) through NORTE 2020 (the North Regional Operational Program 2014/2020). For authors at the CITAB research centre, this research was further financed by the FEDER/COMPETE/POCI—Operational Competitiveness and Internationalization Programme under Project POCI-01-0145-FEDER-006958, and by National Funds of FCT—Portuguese Foundation for Science and Technology under the project UID/AGR/04033/2019. For the author in the CQVR, this research was additionally supported by National Funds of FCT—Portuguese Foundation for Science and Technology under the project UID/QUI/00616/2019.

**Conflicts of Interest:** The authors declare no conflict of interest. The funders had no role in the design of the study; in the collection, analyses, or interpretation of data; in the writing of the manuscript or in the decision to publish the results.

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
