# Peer review of "Hydrologic Impacts of Land Use Changes in the Sabor River Basin: A Historical View and Future Perspectives"

_water, doi:10.3390/w11071464_

Round 1

Reviewer 1 Report

This paper deals with comprehensive water yield changes due to the LULC variations. The authors showed those results using SWAT properly. The wild fire scenario and afforestation projection are quite novel applicatio and interesting. I would recommend this manuscript to be published as it is.

Author Response

Many thanks for the positive appreciation on our manuscript.

Reviewer 2 Report

English language must improvedas some parts are not clearly undrestandable.

Abstract:Please rephrase some parts many times repeat the word used and confuse the reader.

Idroduction: At the end of the introduction the novelty and added value of this research in comparison with other similar studies must be clarified . Why this study is innotative?

At the chapter 2.8. Future scenarios of afforestation and wildfires give some comments if these scenario liked with IPCC or not and why?

Author Response

Answers to Reviewer #2

English language must improve as some parts are not clearly understandable

The English language has been reviewed by the MDPI Editing Services. Moreover, we have revisited the text again for further improvements.

Abstract: Please rephrase some parts many times repeat the word used and confuse the reader.

Yes, in the first phrase the word used was used two times. I did the corrections from L14 and L15. The sentence is the following: “The study area used for this study was the Sabor river basin (located in the northeast of Portugal), which is composed mostly for agroforestry. “

Introduction: At the end of the introduction the novelty and added value of this research in comparison with other similar studies must be clarified. Why this study is innovative?

Yes, I insert the explanation from L116 and L122. The sentence is the following:

A majority of articles discussing environmental effects of land use change and dealing with scenario creation only assess some water balance components, namely water yield or surface flow. This work intends to study the changes occurring not only in water yield or surface flow but also in evapotranspiration, lateral flow and in groundwater flow.

At the chapter 2.8. Future scenarios of afforestation and wildfires give some comments if these scenarios liked with IPCC or not and why?

The scenarios were not based in the IPCC forecast. The afforestation scenario was based on the Regional Plan for Forest Management (RPFM), and the wildfires scenario was based on the burned area occurred in Sabor river basin between 1990 and 2007. The Regional Plan for Forest Management (Portuguese Regulate Decree no. 3/2007, published on 17 January 2007) includes a plan to be accomplished until 2045, and the main objectives are to reduce the risk of wildfire occurrence, and to adjust the proportions of resinous and deciduous species based on the application of correct forest management models. In the RPFM, the Sabor river basin occupies eleven homogeneous sub-regions in the Douro and Northeast regions, which are referenced in the supplementary material (worksheet 2).